# Prevalence, Molecular Characterization and Antimicrobial Susceptibility of *Clostridioides difficile* Isolated from Pig Carcasses and Pork Products in Central Italy

**DOI:** 10.3390/ijerph182111368

**Published:** 2021-10-29

**Authors:** Caterina Licciardi, Sara Primavilla, Rossana Roila, Alessia Lupattelli, Silvana Farneti, Giuliana Blasi, Annalisa Petruzzelli, Ilenia Drigo, Enrico Di Raimo Marrocchi

**Affiliations:** 1Istituto Zooprofilattico Sperimentale dell’Umbria e delle Marche “Togo Rosati”, Via Salvemini 1, 06126 Perugia, Italy; caterina.licciardi@gmail.com (C.L.); a.lupattelli@izsum.it (A.L.); s.farneti@izsum.it (S.F.); g.blasi@izsum.it (G.B.); a.petruzzelli@izsum.it (A.P.); e.diraimo@izsum.it (E.D.R.M.); 2Department of Veterinary Medicine, University of Perugia, Via San Costanzo 4, 06126 Perugia, Italy; 3Istituto Zooprofilattico Sperimentale delle Venezie, Vicolo Mazzini 4, 31020 Treviso, Italy; idrigo@izsvenezie.it

**Keywords:** *Clostridioides difficile*, pig carcasses, pork products, food safety, real-time PCR, ribotype, MALDI-TOF, toxin genes, emerging foodborne pathogens, community-acquired CDI

## Abstract

In the last decade, the incidence and severity of *Clostridioides difficile* infections (CDIs) in humans have been increasing and community-associated infections have been described. For these reasons, the interest in *C. difficile* in food and in food animals has increased, suggesting other possible sources of *C. difficile* acquisition. This study evaluated the presence of *C. difficile* on pig carcasses at the slaughterhouse and in pork products in Central Italy. The contamination rate on pig carcasses was 4/179 (2.3%). Regarding food samples, a total of 216 pork products were tested (74 raw meat preparations and 142 ready-to-eat food samples made by cured raw meat). The real-time PCR screening was positive for 1/74 raw meat preparation (1.35%) and for 1/142 ready-to-eat food samples (0.7%) *C. difficile* was isolated only from the raw meat preparation (pork sausage). All the isolated strains were toxigenic and susceptible to all the tested antibiotics. Strains isolated from carcass samples displayed A+B+CDTa+CDTb+ profile, were toxinotype IV and belonged to the same ribotype arbitrary named TV93, while the one isolated from food samples displayed A+B+CDTa-CDTb- profile and it was not possible to determine ribotype and toxinotype, because it was lost after freeze storage. It was concluded that the prevalence of *C. difficile* in the pork supply chain is very low.

## 1. Introduction

*Clostridioides difficile* (previously known as *Clostridium difficile*) [1] is an anaerobic, spore-forming bacterium and it is considered one of the main causes of nosocomial diarrhoea in hospitalized patients after antibiotic treatment [2].

In the last decade, the incidence and severity of *C. difficile* infections (CDIs) in humans have been increasing due partly to the spread of hypervirulent strains [3]. Recently community-associated infections have been described and CDI is becoming a widespread cause of diarrhea in younger individuals and in populations lacking traditional risk factors, such as hospitalization and antibiotic treatment [4]. Recent data from North America and Europe suggest that 20%–27% of all CDI cases are community-acquired, with an incidence of 20–30 per 100,000 population [5,6,7]. The Italian rate of CDI, including the number of community-acquired outbreaks, is not definitively assessed because of the fragmented nature of the data available in the literature [8].

This increasing incidence has led to the investigation of other possible sources of *C. difficile* acquisition, including the ingestion of contaminated food [9].*C. difficile* has been reported in farm animals [10,11] and in different food products, including those of porcine origin [12,13,14,15]. Recent studies have focused on pigs and cattle as *C. difficile* carriers, suggesting the possibility of carcass contamination during the slaughter process and a potential food safety issue [16,17,18]. Data describing the prevalence of *C. difficile* on pig carcasses at slaughterhouses show significant differences related to the geographical contest. Concerning the European background, a Belgian study reported a prevalence of 7% [18], while, in other world countries, in particular in America and Asia, the contamination rate is higher and it ranges from 8% to 30% [16,17,19,20,21].

Among food samples *C. difficile* was recently isolated from a variety of meat products, including pork products such as: pork sausages [15,16,22,23], ground pork samples [14,15,21,23,24] and ready-to-eat pork products [23], reinforcing the hypothesis of an emerging foodborne pathogen.

PCR-ribotype 078 (RT-078) has been commonly isolated in food products and food animals and some authors consider this ribotypeas potentially involved in foodborne transmission to humans [23,25,26,27,28,29]. This ribotypeis often found in community-acquired infections and it is among the ten most frequently isolated ribotypes in European populations [30]. RT-078 is the most common ribotype in pigs, cattle and horses worldwide [31] and in a recent study performed in Central Italy, it was detected in dairy and beef cattle farms [32]. In the same study, RT-126 turned out to be the most commonly isolated ribotype in calves [32]. The prevalence of this ribotype in humans is variable, ranging from 3% in a European survey [33] to 34.4% in Spain [34]. In a different study conducted, in hospitals of Central Italy, this ribotype was also recovered from a hospital food (lettuce) and it was the second most frequently isolated ribotype between human cases of CDI recorded in the same facility [35].

The aim of this study was to evaluate the presence of *C. difficile* on pig carcasses at the slaughterhouse and in pork products in Central Italy. *C. difficile* isolates were characterized and compared to the main PCR-ribotypes circulating in the same area.

## 2. Materials and Methods

### 2.1. Samples Collection

Between January 2019 and September 2020, 179 carcass samples have been collected from 5 pig slaughterhouses located in Central Italy (Umbria and Marche regions). In particular, three slaughterhouses were classified as small-sized (capacity of <5000 animals per year) and 2 as medium-sized (capacity of >5000 to <50000 animals per year). The animal sample was calculated to estimate prevalence with 95% confidence interval (CI) and 5% desired absolute precision, considering a 15% expected prevalence, on the basis of previous investigations [18,19]. Four different points (ham, basin, sternum and forelimb) were swabbed using a single hydrated sponge (Solar-cult Pre-moistened Sponges—Solar Biologicals Inc, Ogdensburg, NY, USA) [18], the samples were stored at 4 °C and processed within 24–48 h. This carcass sampling method was defined according to thatpreviously reported by Rodriguez et al., 2013 [18].

Pork products were collected from samples submitted to our laboratories for routine diagnosis. They included raw meat preparations (pork sausages) and ready-to-eat food samples (salami, spreadable salami, cured sausages, bacon, cured pork cheek, cured ham, cured pork shoulder and cured pork lean sirloin).

### 2.2. Enrichment and Real-Time PCR Screening

Carcass sponges were put into 50 mL of Taurocholate Cefoxitin Cycloserine Fructose Broth (TCCFB) (Ethanol 96%—Sigma-Aldrich Corporation, St. Louis, MO, USA); *C. difficile* selective supplement—Oxoid Limited, Basingstoke, UK; D-fructose—Sigma-Aldrich Corporation, St. Louis, MO, USA; monobasic potassium phosphate—Sigma-Aldrich Corporation, St. Louis, MO, USA; proteose peptone—Biolife Italiana s.r.l., Milan, IT; neutral red—Sigma-Aldrich Corporation, St. Louis, MO, USA; sodium phosphate dibasic—Chem-lab, Zedelgem, Belgium; sodium taurocholate hydrate—Sigma-Aldrich Corporation, St. Louis, MO, USA) and incubated in anaerobic jars (2.5 L AnaeroJar, AG0025 with AnaeroGen 2.5 L, AN0025, Oxoid Limited, Basingstoke, UK) at 37 °C for 12 days [18].

Ten grams of each food sample was added to 90 mL of *C. difficile* Moxalactam Norfloxacin Broth (CDMNB) (CDMN selective supplement—Oxoid Limited, Basingstoke, UK; D-fructose—Sigma-Aldrich Corporation, St. Louis, MO, USA; magnesium sulfate heptahydrate—Merck Millipore, Burlington, MA, USA; monobasic potassium phosphate—Sigma-Aldrich Corporation, St. Louis, MO, USA; proteose peptone—Biolife Italiana srl, Milan, Italy; defibrinated horse blood—Allevamento Blood di Fiastra Maddalena, Teramo, Italy; sodium chloride—Sigma-Aldrich Corporation, St. Louis, MO, USA; sodium phosphate dibasic—Chem-lab, Zedelgem, Belgium; sodium taurocholate hydrate—Sigma-Aldrich Corporation, St. Louis, MO, USA) and incubated in anaerobic jars at 37 °C for 12 days [36].

After the incubation, 1 mL of each sample was used for DNA extraction using 6% Chelex-100 sodium form (Sigma-Aldrich Corporation, St. Louis, MO, USA).

In order to detect *C. difficile*, a real-time PCR screening was performed, amplifying a species-specific internal fragment of the triose phosphate isomerase (*tpi*) housekeeping gene, using the primers *tpi-F* [5′-AAAGAAGCTACTAAGGGTACAAA-3′] and *tpi-R* [5′-CATAATATTGGGTCTATTCCTAC-3′], described by Lemee et al. [37]. The PCR reaction was carried out in 20 µL of final mix containing: 2 µL of DNA, 10 µL of a ready-to-use mix of Taq polymerase and SYBR Green (KAPA SYBR FAST qPCR Master Mix (2X) Universal; KK 4601, Kapa Biosystems, Wilmington, MA, USA), 450 nM of each primer, forward and reverse, 1 nM of ROX and water for molecular biology.Amplification was performed on Stratagene Mx3005P instrument (Agilent Technologies, Santa Clara, CA, USA) under the following conditions, as described by Morales et al. [38] and according to the Master Mix manufacturer’s instructions: initial denaturation at 95 °C for 3 min, 40 cycles at 95 °C for 3 s and 60 °C for 30 s, with a final melting curve ranging from 65 °C to 95 °C. A negative control, consisting of water for molecular biology and a positive control, represented by a reference *C. difficile* strain (CDC20120296—Microbiologics, St. Cloud, MN, USA) were also set up. Under these conditions, the melting temperature of the *tpi* amplicon was 78 °C, whereas that of the primer dimers was 71 °C.

### 2.3. Isolation and Identification

The positive broth cultures were alcohol shocked by mixing 2 mL of broth with 96% ethanol (1:1 *v*/*v*) (Sigma-Aldrich Corporation, St. Louis, MO, USA) for 30 min at room temperature. After centrifugation (3800× *g* for 10 min) the sediment was streaked onto Taurocholate Cefoxitin Cycloserine Fructose Agar (TCCFA) (TCCFB additioned with agar—Biolife Italiana s.r.l., Milan, Italy) for carcass sample [18] and *C. difficile* Moxalactam Norfloxacin Agar (CDMNA) (CDMN selective supplement—Oxoid Limited, Basingstoke, UK; *C. difficile* agar base—Oxoid Limited, Basingstoke, UK; defibrinated horse blood—Allevamento Blood di Fiastra Maddalena, Teramo, Italy) for food samples [36] and incubated for 48 h in anaerobic jars. Suspected colonies (rhizoid colonies, non-hemolytic) were presumptively identified on the basis of Gram and latex agglutination test (*C. difficile* test kit—Oxoid Limited, Basingstoke, UK) and then confirmed using a MALDI-TOF MS instrument (Bruker Daltonics, Bremen, Germany) with Microflex LT Smart Biotyper and Flex Control Biotyper 3.4 software (Bruker Daltonics, Bremen, Germany). Briefly, the formic acid extraction technique was performed for each isolate by adding one or two colonies to a microcentrifuge tube containing 300 μL of HPLC grade deionized water. The suspension was mixed thoroughly by pipetting, and 900 μL of absolute ethanol was further added and mixed thoroughly again. After a centrifugation of 2 min at 3000× *g*, the supernatant was discarded and 50 μL of 70% formic acid was added to the pellet and vortexed. Fifty μL f 100% acetonitrile was added and mixed, followed by a 2 min centrifugation at 3000× *g*. One μL of the supernatant was added to the target slide and dried at room temperature. One μL of the HCCA matrix was applied on the sample spot and dried at room temperature.

### 2.4. Strains Characterization

All the confirmed strains were screened by two different multiplex end-point PCRs: one for *tpi* and the toxin genes *tcdA* and *tcdB* [37] and the other for the binary toxin genes *cdtA* and *cdtB* [39], using the primers described in literature.

DNA extraction was performed on 48-h blood agar cultures, by boiling at 100 °C for 5 min and subsequently frozen at −20 °C for 10 min, the mixture was then centrifuged at 3000× *g* for 10 min [40].

The PCR reactions were performed on Eppendorf Mastercycler instrument (Eppendorf s.r.l., Milan, Italy) according to the conditions given by Lamee et al., 2004 [37], Doosti et al., 2014 [39] and to the Master Mix manufacturer’s instructions. The first multiplex PCR reaction (*tcdA*, *tcdB* and *tpi*) [37] was carried out in 50 µL of final mixture containing: 2 µL of DNA, 25 µL of HotStarTaqMM 2X (Qiagen, Hilden, Germany) 2 mM MgCl_2_ (Qiagen, Hilden, Germany), 0.5 μM of *tpiF* and *tpiR* primers and 1 μM of the other primers. The conditions used were: initial denaturation at 95 °C for 15 min, 11 cycles at 95 °C for 30 s, 65–55 °C (scaling one degree each cycle) for 30 s, 72 °C for 30 s, 29 cycles at 95 °C for 30 s, 55 °C for 30 s, 72 °C for 30 s and a final extension step at 72 °C for 5 min. The second one (*cdtA* and *cdtB*) [28] was carried out in 50 µL of final mix containing: 2 µL of DNA, 25 µL of HotStarTaqMM 2X (Qiagen, Hilden, Germany), 6mM MgCl_2_ (Qiagen, Hilden, Germany), 1 μM of each primer. The reaction was performed under the following conditions: initial denaturation at 95 °C for 15 min, 40 cycles at 94 °C for 60 s, 59 °C for 60 s, 72 °C for 60 s and a final extension step at 72 °C for 7 min. To check each PCR session a negative control, consisting of water for molecular biology and a positive control, represented by a reference *C. difficile* strain (CDC20120296—Microbiologics, St. Cloud, MN, USA) were set up. The PCR products were uploaded in the QIAxcel System Instrument (Qiagen, Hilden, Germany), an automated capillary electrophoresis device and analyzed by the QIAxcelScreengel 1.4.0 software (Qiagen, Hilden, Germany) for size determination of the detected fragments.

Toxinotyping and PCR-ribotyping were performed according to the protocol described by Rupnik et al. [41,42] and Bidet et al. [43], respectively. For PCR-ribotyping, the profile of the testes strains were compared to the predominant PCR-ribotypes circulating in Europe (RT-001, RT-002, RT-003, RT-005, RT-010, RT-012, RT-016, RT-017, RT-018, RT-014/020, RT-027, RT-031/1, RT-033, RT-050, RT-056, RT-070, RT-078, RT-081, RT-103, RT-126, RT-127, RT-150, RT-205, RT-403, RT-439, RT-449, RT-548, RT-592, RT-614). Isolates not showing any matches were named using an internal nomenclature (Treviso, TV and number).

### 2.5. Antimicrobial Susceptibility Testing

Minimum inhibitory concentrations (MICs) were determined for clindamycin (CLI), metronidazole (MTZ), moxifloxacin (MXF) and vancomycin (VAN) using Etest strips (CLI,509518—MTZ,530018—MXF,529018—VAN,525518—bioMèrieux, Marcy-L’Étoile, France), following the manufacturer’s technical guide. Bacterial suspensions were streaked on Brucella Blood Agar (Brucella Agar—Becton Dickinson, Franklin Lakes, NJ, USA; Hemin—Sigma-Aldrich Corporation, St. Louis, MO, USA; defibrinated horse blood—Allevamento Blood di Fiastra Maddalena, Teramo, Italy) and incubated for 48 h in anaerobic jars. MICs were recorded after the incubation and analyzed according to the epidemiological cut-off (ECOFF) (EUCAST, https://www.eucast.org, accessed on 15 July 2020). Isolates with MIC values >2 mg/L for MTZ and VAN, >4 mg/L for MXF and >16 mg/L for CLI were considered with reduced antimicrobial susceptibility. Furthermore, *Bacteroides fragilis* ATCC 25285 was included as a quality control strain.

## 3. Results

### 3.1. Prevalence of C. Difficile in Carcass Samples and Food Samples

A total of 179 pig carcass samples were analyzed, PCR screening was positive for 4/179 samples (2.3%) and *C. difficile* was isolated from all the positive ones. The isolates obtained were called: CDS1, CDS2, CDS3, CDS4.

Concerning food samples, a total of 216 pork products were tested. They included 74 raw meat preparations (pork sausages) and 142 ready-to-eat food samples made by cured raw meat. The real-time PCR screening was positive for 1/74 raw meat preparation (1.35%) and for 1/142 ready-to-eat food samples (0.7%) *C. difficile* was isolated only from the raw meat preparation (pork sausage) and the isolate was called CDF1. Strain isolation from the ready-to-eat food sample (salami) resulted negative, probably due to the low level of contamination or to the presence of the bacterium in a non-viable form (Table 1).

The colonies were identified as *C. difficile* by MALDI-TOF MS. Except for CDF1, all the isolates had a high-confidence identification spectral score (Table 2) and high consistency interpretation (A), since the same species was identified as the second- and third-best match. The strains were further confirmed through molecular investigations.

### 3.2. Strains Characterization and Antimicrobial Susceptibility

All the isolated strains were confirmed as *C. difficile* due to the presence of *tpi* (Figure 1). CDS1, CDS2, CDS3, CDS4 (isolated from carcass sponges) belonged to PCR-ribotype TV93, toxinotype IV and displayed A^+^B^+^CDTa^+^CDTb^+^ profile (Figure 1, Table 3), CDF1 (isolated from pork sausages) displayed A^+^B^+^CDTa^-^CDTb^-^ profile (Figure 1, Table 3), while ribotype and toxinotype determination was not possible, because it was lost after the storage procedure. The isolates were susceptible to all the antibiotics tested (Table 3).

## 4. Discussion

The present study determined the prevalence of *C. difficile* in pig carcasses and in pork products in Central Italy.

*C. difficile* was recovered from four carcasses (4 out of 179—2.2%) and all the positive samples came from the same slaughterhouse (Umbria region). The isolated strains belonged to a ribotype that did not match the reference strains available in the laboratory, nor with the human strains tested in previous studies conducted in the same area [35]. Our results agree with what was observed in a previous European study on pig carcasses [18], while it significantly differs from data obtained in other countries [16,17,19], where higher prevalence is reported (15–30%). This suggests that in Europe the prevalence of *C. difficile* in pig carcasses at slaughterhouses is very low.

Molecular screening of food samples showed a total prevalence of *C. difficile* of 0.9% (2 out of 216), in particular: 1.4% (1 out of 74) for raw products to be consumed cooked and 0.7% (1 out of 142) for ready-to-eat products. The only positive sample, from which it was possible to isolate *C. difficile*, was the raw meat preparation (pork sausages). In the cured meat product (salami), isolation was not possible, probably due to a very low concentration of the microorganism or to the presence of the germ in a no longer viable form. As regards pork sausages, the prevalence we observed was slightly lower than a previous European study (5 of 107 uncooked pork samples—4.7%) [15] and an American study, where the prevalence of *C. difficile* in pork sausages was 4.9% (2 of 41) [22]. Almost all the bibliographic data, about food of swine origin, concerns raw products to be consumed cooked and not ready-to-eat products. An American study, with a lower sampling size, detected higher prevalence (from 14.3% to 62.5%) than what we observed in different kinds of ready-to-eat pork products [23]. Some studies, instead, completely failed to detect *C. difficile* in pork and pork products in The Netherlands, Switzerland, France, and Sweden [44,45,46,47]. Differences in prevalence observed between our study and previous studies could be due to different sampling methods and isolation conditions, making it difficult to compare results and to give an objective evidence for foodborne transmission.As already reported for *Salmonella spp.* in pig carcasses [48], a harmonized and integrated approach along the food chain and across different countries would be desirable to reduce the presence of these pathogens in pork and pork products.

As regards antimicrobial susceptibility, in our study, the isolates were fully susceptible to all the antimicrobials tested, as already observed by Rodriguez el al. [15] in strains not associated with any reference ribotypes. In pork meat, MXF resistance has been previously reported for *C. difficile* isolates belonging to PCR-ribotype 078 [15]. In a prospective study of CDIs in Europe, the same resistance was observed in 37.5% of the clinical strains circulating in hospitals [49]. Low susceptibility to CLI in *C. difficile* pork meat isolates belonging to PCR-ribotype 027 has been observed [23], and this may not be due to the widespread use of tylosin, erythromycin, virginiamycin, and lincomycin in food animals and the consequent potential for selection of macrolide-lincosamide-streptogramin resistance [50]. In human therapy, VAN together with MTZ are the recommended first-line antimicrobials for the treatment of CDIs [51], although one of the most important risk factors for the disease is exposure to antibiotics [52,53] and there is an increasing concern about the emergence of antibiotic resistance resulting in treatment failure [54]. A valuable alternative could be potentially represented by the use of natural antimicrobials, especially those extracted from food industry by-products, along the food production chain to inhibit or limit microbial growth [55,56,57,58]. This approach would allow for the pursuing of both public health protection and environmental sustainability.

In summary, our data suggest that *C. difficile* occurs as a low-level contaminant in the pork supply chain. Despite the low number of isolated strains, all of them carried virulence genes capable of causing diarrhea and colitis in humans, in particular in vulnerable groups that should be advised to avoid potentially contaminated products. Additional studies are needed to characterize risks posed by this organism in the human food supply and its clinical relevance.

## Figures and Tables

**Figure 1 ijerph-18-11368-f001:**
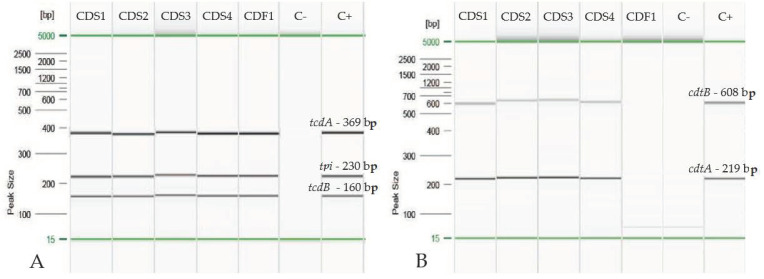
Capillary electrophoresis results (QIAxcelScreengel 1.4.0 software). Multiplex PCR for detection of *tpi*, *tcdA* and *tcdB* (**A**). Multiplex PCR for detection of *cdtA* and *cdtB* (**B**).

**Table 1 ijerph-18-11368-t001:** Prevalence of *C. difficile* in pork products (raw meat preparations and ready-to-eat products).

Sample Types	No. Tested Samples	No. (%) Positive Samples (Real-Time PCR)	No. (%) *C. difficile* Isolates
Sausages	74	1 (1.4%)	1 (1.4%)
Total raw meat preparations	74	1 (1.4%)	1 (1.4%)
Salami	64	1 (1.6%)	0 (0%)
Spreadable salami	27	0 (0%)	0 (0%)
Cured sausage	25	0 (0%)	0 (0%)
Bacon	7	0 (0%)	0 (0%)
Cured pork’s cheek	1	0 (0%)	0 (0%)
Cured ham	10	0 (0%)	0 (0%)
Cured pork’s shoulder	4	0 (0%)	0 (0%)
Cured pork’s lean sirloin	4	0 (0%)	0 (0%)
Total ready-to-eat pork products	142	1 (0.7%)	0 (0%)

**Table 2 ijerph-18-11368-t002:** Identification spectral score of MALDI-TOF MS of *C. difficile*.

Isolate ID	Score
CDS1	2.18
CDS2	2.23
CDS3	2.16
CDS4	2.19
CDF1	1.90

**Table 3 ijerph-18-11368-t003:** Ribotype, toxinotype and MICs values (mg/L) of *C. difficile* isolates.

Isolate ID	Source of isolation	Ribotype	Toxinotype	Detection of Toxin Genes by PCR				Antimicrobials ^1^			
				tcdA	tcdB	cdtA	cdtB	CLI	MTZ	VAN	MXF
CDS1	Carcass sponge	TV93 ^2^	IV	+	+	+	+	3	0.25	1	0.5
CDS2	Carcass sponge	TV93 ^2^	IV	+	+	+	+	1	0.125	0.75	0.38
CDS3	Carcass sponge	TV93 ^2^	IV	+	+	+	+	2	0.19	0.75	0.5
CDS4	Carcass sponge	TV93 ^2^	IV	+	+	+	+	1	0.125	1	0.38
CDF1	Pork sausage	NT ^3^	NT ^3^	+	+	-	-	2	0.19	0.75	0.5

^1^ CLI = Clindamycin, MXF = Moxifloxacin, MTZ = Metronidazole, VAN = Vancomycin. ^2^ TV stands for internal nomenclature (Treviso, TV and number). ^3^ NT= not tested.

## Data Availability

Data are available from the authors.

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
