# Peer review of "Prevalence, Molecular Characterization and Antimicrobial Susceptibility of Clostridioides difficile Isolated from Pig Carcasses and Pork Products in Central Italy"

_ijerph, 2021, doi:10.3390/ijerph182111368_

Round 1

Reviewer 1 Report

The authors describe a study for the detection of Clostridioides difficile from Pig Carcasses and Pork Products in Central Italy.

The paper needs revising in order to have a publishing standard. (The inclusion of line numbers will help the reviewing process...)

Introduction

1.“C. difficile has been reported in farm animals [5,6] and in different kinds of food, including pork products [7-10] and PCR-ribotype 078 results predominant in food and food animals, suggesting that this ribotype has the potential of foodborne transmission to humans [11-16].”

The sentence doesn’t make sense. Please revise

2.“In the USA, Harvey et al. indicated a prevalence of 30% in Texas [17], in North America the prevalence detected ranged from 8 to 20% [21,22], while in Canada the prevalence of C. difficile in slaughterhouses was 15% [18].”

Texas is in North America. Revise sentence

Materials and Methods : Sample collection

  1. “…with a 15% expected prevalence, 95% CI and 5% precision, on the basis of previous investigation”

This sentence doesn’t make sense in this part of the manuscript

  1. “…using the primers described by Lemee et al. [26]”

Please describe primers and PCR conditions.

  1. “To perform and interpret this real-time PCR the indications given by Morales et al. were used [27].”

That kind of interpretation was performed please give a detailed explanation

  1. Strains characterization

DNA extraction was performed on 48-hours blood agar cultures, by boiling at 100°C

for 5 min and subsequently frozen at -20°C for 10 min…”

Why the authors didn’t use always the same extraction method? Please explain

  1. Discussion

“and researchers have started to look for new therapeutic options for CDI, including the use of natural antimicrobials of plant origin [44]. This kind of approach can be also applied to food production in order to replace the use of chemical compounds to inhibit or limit microbial growth as already assessed both for pathogenic and spoilage microorganisms [45-47]. A precedent study, highlighted the antimicrobial activity of a number of natural products against C. difficile in vitro with less effect against”

The work is not about the new compounds. This part of the discussion is not necessary

In general, the authors could better explain the results obtained

Author Response

Answer to Reviewer 1

Dear Editor of International Journal of Environmental Research and Public Health,

we are pleased to submit the revised version of our manuscript.

Please find hereafter a comprehensive list of the comments to the observations of the reviewer 1. All referees’ comments were addressed and corrections were made.

All changes are marked up using the “Track Changes” function in the revised manuscript.

We hope that these revisions improve the paper such that you and the reviewer now deem it worthy of publication in the Journal.

The authors would like to thank the reviewer 1 for the thorough revision of the manuscript.

Point 1:C. difficile has been reported in farm animals [5,6] and in different kinds of food, including pork products [7-10] and PCR-ribotype 078 results predominant in food and food animals, suggesting that this ribotype has the potential of foodborne transmission to humans [11-16].”

The sentence doesn’t make sense. Please revise

Response 1: Thank you for the comment. The sentence was rephrased as suggested and further argued later in the introduction section.

Point 2:“In the USA, Harvey et al. indicated a prevalence of 30% in Texas [17], in North America the prevalence detected ranged from 8 to 20% [21,22], while in Canada the prevalence of C. difficile in slaughterhouses was 15% [18].”

Texas is in North America. Revise sentence

Response 2: Thank you for the comment. The sentence was rephrased.

Point 3: Materials and Methods: Sample collection. “…with a 15% expected prevalence, 95% CI and 5% precision, on the basis of previous investigation” This sentence doesn’t make sense in this part of the manuscript.

Response 3: Thank you for the comment. The sentence was rephrased to make it more suitable for this part of the manuscript.

Point 4: “…using the primers described by Lemee et al. [26]”. Please describe primers and PCR conditions.

Response 4: Thank you for the comment. The primers sequence and the PCR conditions were added. The same kind of details were added also for the two end point PCRs performed for strain characterization.

Point 5:  “To perform and interpret this real-time PCR the indications given by Morales et al. were used [27].” That kind of interpretation was performed please give a detailed explanation

Response 5: Thank you for the comment. Aiming to make the interpretation of real-time PCR results clearer to the reader, the authors detailed the parameters used to interpret the real-time PCR (melting temperature of positive samples) as suggested.

Point 6: Strains characterization DNA extraction was performed on 48-hours blood agar cultures, by boiling at 100°C for 5 min and subsequently frozen at -20°C for 10 min…”

Why the authors didn’t use always the same extraction method? Please explain

Response 6: Thank you for the comment. The authors used two different extraction methods due to the different characteristics of matrices subjected to analysis: Chelex-100 resin method for broth cultures screening and boiling method to apply to isolated strains characterization, as already performed in previous investigations (Primavilla et al. 2019). The use of Chelex-100 resin method for broth cultures screenings has been also described for Clostridium botulinum detection as a faster and cost effective method compared to other extractions methods (Auricchio et al. 2013) and suitable for screening investigations of such impure matrix. The boiling method, instead, is commonly employed for pure cultures extraction and it has been already described for confirmation of C. difficile isolates (Marcos et al, 2021).   

Point 7: Discussion “and researchers have started to look for new therapeutic options for CDI, including the use of natural antimicrobials of plant origin [44]. This kind of approach can be also applied to food production in order to replace the use of chemical compounds to inhibit or limit microbial growth as already assessed both for pathogenic and spoilage microorganisms [45-47]. A precedent study, highlighted the antimicrobial activity of a number of natural products against C. difficile in vitro with less effect against” The work is not about the new compounds. This part of the discussion is not necessary.

Response 7: Thank you for the comment. The authors agree with the reviewer that the study does not focus on new compounds. However, the antimicrobial susceptibility of C. difficile represents one of the major aspects investigated and reported in the manuscript. According to the reviewer's suggestion this part of the discussion was streamlined and rephrased. Unless contrary reviewer's opinion, the authors would really much rather keep a few lines regarding the potential use of natural antimicrobials to contrast the antimicrobial resistance concern, as this brief comment is, in our opinion, suitable for the discussion, in the topic and in line with Journal purpose.

Point 8: In general, the authors could better explain the results obtained

Response 8: Thank you for the comment. As suggested by the reviewer, the results section was further discussed. More details corroborating PCR results and MALDI-TOF data were added in terms of additional information and clarificatory images

Reviewer 2 Report

Dear Authors,

here I provide some comments needed to be taken into account.

  1. Last paragraph of introduction: "The aim of this study was to evaluate the presence of C. difficile on pig carcasses at the slaughterhouse and in pork products in Central Italy. C. difficile isolates were characterized and compared to the main PCR-ribotypes found in humans in the same area."

Please give a more general information in this section about PCR-ribotypes of C. difficile.

  1. „In the last decade, the incidence and severity of C. difficile infections (CDIs) in humans have been increasing, also because of the spread of hypervirulent strains [3]. Recently community-associated infections have been described, and the interest in C. difficile in food and in food animals has increased, suggesting other possible sources of C.difficile acquisition [4].”

Please give a particular data, I mean epidemiological data concerning number of cases in humans worldwide and their symptoms.

  1. Materials and Methods 1. Samples collection: „Four different points (ham, basin, sternum and forelimb) were swabbed using a single hydrated sponge (Solar-cult Pre-moistened Sponges - Solar Biologicals Inc, Ogdensburg, NY, USA) [19], the samples were stored at 4°C and processed within 24-48 h.

Is there some reason to collect probes from these points? If yes, just justify why did you choose this points for swabbing. In some papers concerning the bacteriological contamination of pork carcasses you can find other parts of swabbing, i.e. edible giblet samples, intenstines, lymph nodes or tonsils. Does the swab site matter for C. difficile detection? Or, would you expect higher C. difficile detectability from another part of the carcass? Is there any known way of spreading of C. difficile within pork carcasses? Please explain and add apropriate citation.

  1. Isolation and identification by MALDI-TOF mass spectrometry (MALDI Biotyper, Bruker Daltonics, Billerica, MA, USA).

Please give some references for that method, give some results from that identification: mass spectra generated are analyzed by appropriate software and compared with profiles gathered in the database (it means score value). It could by even in the supplementary.

  1. 4. Strains characterization „All isolates, included in this study, were screened by end-point PCRs for the toxin genes tcdA, tcdB and the binary toxin genes cdtA and cdtB”. Or 3.2. Toxin genes detection, PCR-ribotyping toxinotyping and antimicrobial susceptibility.

Please show results from that screening. Do you have any electrophoretic gels with positive controls to proof the presence of toxin genes? It could by even in the supplementary.

  1. Table 2. Ribotype, toxinotype and MICs values (mg/L) of difficile isolates.

Please add the source of isolation in an additional column.

  1. „Samples collection: Four different points (ham, basin, sternum and forelimb) were swabbed using a single hydrated sponge”

Could you indicate in table 2 the exact isolation site of CDS1, CDS2, CDS3, CDS4? I did not find it in the text and it seems to be an interesting information.

  1. General comment: to many „As regards” used along the paper.

  2. Please give in a nutshell some epidemiological data, i.e. number of human infection caused by difficile in Italy and Europe based on current ECDC data, in introduction or discussion.

  3. „Additional studies are needed to characterize risks posed by this organism in the human food supply and its clinical relevance.”

Are you able to give some examples of such research?

Author Response

Answer to Reviewer 2

Dear Editor of International Journal of Environmental Research and Public Health,

we are pleased to submit the revised version of our manuscript.

Please find hereafter a comprehensive list of the comments to the observations of the reviewer 2. All referees’ comments were addressed and corrections were made.

All changes are marked up using the “Track Changes” function in the revised manuscript.

We hope that these revisions improve the paper such that you and the reviewer now deem it worthy of publication in the Journal.

The authors would like to thank the reviewer 2 for the thorough revision of the manuscript.

Point 1: Last paragraph of introduction: "The aim of this study was to evaluate the presence of C. difficile on pig carcasses at the slaughterhouse and in pork products in Central Italy. C. difficile isolates were characterized and compared to the main PCR-ribotypes found in humans in the same area."

Please give a more general information in this section about PCR-ribotypes of C. difficile.

Response 1: Thank you for the comment. As suggested, the authors added in the introduction information about the most frequently isolated PCR-ribotypes, with particular emphasis on data collected in the same area covered by this study.

Point 2: In the last decade, the incidence and severity of C. difficile infections (CDIs) in humans have been increasing, also because of the spread of hypervirulent strains [3]. Recently community-associated infections have been described, and the interest in C. difficile in food and in food animals has increased, suggesting other possible sources of C.difficile acquisition [4].”

Please give a particular data, I mean epidemiological data concerning number of cases in humans worldwide and their symptoms.

Response 2: Thank you for the comment. According to reviewer suggestion, the authors gave epidemiological data in Europe and America, describing the increasing incidence of community-acquired CDI.

Point 3: Materials and Methods 1. Samples collection: „Four different points (ham, basin, sternum and forelimb) were swabbed using a single hydrated sponge (Solar-cult Pre-moistened Sponges - Solar Biologicals Inc, Ogdensburg, NY, USA) [19], the samples were stored at 4°C and processed within 24-48 h.

Is there some reason to collect probes from these points? If yes, just justify why did you choose this points for swabbing. In some papers concerning the bacteriological contamination of pork carcasses you can find other parts of swabbing, i.e. edible giblet samples, intenstines, lymph nodes or tonsils. Does the swab site matter for C. difficile detection? Or, would you expect higher C. difficile detectability from another part of the carcass? Is there any known way of spreading of C. difficile within pork carcasses? Please explain and add apropriate citation.

Response 3: Thank you for the comment. The authors chose these points for swabbing as already described in previous studies, concerning pigs carcasses contamination with C. difficile (Rodriguez at al. 2013). The reference corroborating this approach was clearly reported as suggested.

Point 4: Isolation and identification by MALDI-TOF mass spectrometry (MALDI Biotyper, Bruker Daltonics, Billerica, MA, USA).

Please give some references for that method, give some results from that identification: mass spectra generated are analyzed by appropriate software and compared with profiles gathered in the database (it means score value). It could by even in the supplementary.

Response 4: Thank you for the comment. As suggested, we described the identification protocol, which was used according to the manufacturer’s instructions. Moreover, we included in the text the interpretation of the MALDI-TOF score values including an additional table in the text (Table 2). The database used for the identification was the MALDI Biotyper released by Bruker, without any changes.

Point 5: Strains characterization „All isolates, included in this study, were screened by end-point PCRs for the toxin genes tcdAtcdB and the binary toxin genes cdtA and cdtB”. Or 3.2. Toxin genes detection, PCR-ribotyping toxinotyping and antimicrobial susceptibility.

Please show results from that screening. Do you have any electrophoretic gels with positive controls to proof the presence of toxin genes? It could by even in the supplementary.

Response 5: Thank you for the comment. According to the reviewer comment, the authors added along the manuscript a detailed description of the end-point PCRs for the toxin genes and an image (Figure 1) to show the results obtained. 

Point 6: Table 2. Ribotype, toxinotype and MICs values (mg/L) of difficile isolates. Please add the source of isolation in an additional column.

Response 6: Thank you for the comment. As suggested, the column was added.

Point 7: Samples collection: Four different points (ham, basin, sternum and forelimb) were swabbed using a single hydrated sponge”

Could you indicate in table 2 the exact isolation site of CDS1, CDS2, CDS3, CDS4? I did not find it in the text and it seems to be an interesting information.

Response 7: Thank you for the comment. As indicated in paragraph 2.1, the four different points were swabbed with a single hydrated sponge and represent a single sample. CDS1, CDS2, CDS3, CDS4 are only the names of the strains isolated from the four positive samples. Therefore, direct relation between the name of the isolate and the specific isolation site is unfeasible.

Point 8: General comment: to many „As regards” used along the paper.

Response 8: Thank you for the comment. Some of the “as regards” were removed, in order to allow for an easy reading.

Point 9: Please give in a nutshell some epidemiological data, i.e. number of human infection caused by difficile in Italy and Europe based on current ECDC data, in introduction or discussion.

Response 9: Thank you for the comment. The authors added, in the introduction section, epidemiological data about the increasing incidence of community-acquired Clostridium difficile infections in America, Europe and Italy. Those data were taken from recent articles, because the last ECDC Annual Epidemiological Report (2016) focused on Healthcare-associated infections.

Point 10: Additional studies are needed to characterize risks posed by this organism in the human food supply and its clinical relevance.”

Are you able to give some examples of such research?

Response 10: Thank you for the comment. As mentioned along the manuscript, several aspects of the investigation were uneasy to discuss and to compare with those in the previous literature, due to high level of variability in terms of the choice of sampling sites, of growth culture media and analytical methods, but also in terms of prevalence registered and ribotype detected. Therefore, in the authors' opinion, further investigation would be useful to collect more data and to harmonize the approach and the techniques used in the detection of C.difficile in the pork production chain in order to elaborate comparable results and to pose the basis for a better understanding of this food safety issue. 

Round 2

Reviewer 2 Report

Dear authors,

Thanks for the answers. I accept the paper in its current form

This manuscript is a resubmission of an earlier submission. The following is a list of the peer review reports and author responses from that submission.